# Body Mass Index and Blood Pressure-to-Height Ratio in Predicting Incidence of Hypertension in Serbian Children

**DOI:** 10.3390/children7120254

**Published:** 2020-11-25

**Authors:** Valerija Puškaš, Rada Rakić, Maja Batez, Dejan Sakač, Tatjana Pavlica

**Affiliations:** 1Department of Biology and Ecology, Faculty of Sciences, University of Novi Sad, 21000 Novi Sad, Serbia; puskasv@eunet.rs (V.P.); rada.rakic@dbe.uns.ac.rs (R.R.); tatjana.pavlica@dbe.uns.ac.rs (T.P.); 2Faculty of Sport and Physical Education, University of Novi Sad, 21000 Novi Sad, Serbia; 3Clinic of Cardiology, Institute of Cardiovascular Diseases of Vojvodina, 21000 Novi Sad, Serbia; dejansakac@gmail.com; 4Medical Faculty, University of Novi Sad, 21000 Novi Sad, Serbia

**Keywords:** diagnosis, elevated blood pressure, indicator, children

## Abstract

Background: A new method using blood pressure-to-height ratio for diagnosing elevated blood pressure/hypertension in children has been introduced recently. We aimed to compare blood pressure-to-height ratio (BPHR) and Body Mass Index (BMI) in predicting incidence of hypertension (HTN). Methods: The sample consisted of 1133 boys and 1154 girls aged 7–15. We used the following equations for BPHR: systolic BPHR (SBPHR) = SBP (mm Hg)/height (cm) and diastolic BPHR (DBPHR) = DBP (mm Hg)/height (cm). In order to determine the accuracy of SBPHR, DBPHR and BMI as diagnostic tests for elevated blood pressure (elevated BP), we used the receiveroperating characteristic curve analyses. Results: The area under the curve (AUC) values for BMI ranged from 0.625 to 0.723 with quite low sensitivity rates from 62% to 72.5% and specificities from 58.2% to 67.3% showing a modest ability to identify children with elevated BP and HTN. On the contrary, BPHR showed a great predictive ability to identify elevated BP and HTN with AUC values of 0.836 to 0.949 for SBP and from 0.777 to 0.904 for DBP. Furthermore, the sensitivity ranged from 78.5% to 95.7%, and the specificity from 73.9% to 87.6%. Conclusion: the current study showed that BPHR is an accurate index for detecting elevated BP and HTN in children aged 7 to 15 years and can be used for early screening.

## 1. Introduction

Hypertension/high blood pressure (HTN/HBP) can be defined as systolic BP (SBP) and/or diastolic BP (DBP) above the 95th percentile, and blood pressure (BP) between the 90th and 95th percentile can be considered as “elevated blood pressure” [1]. Due to high prevalence, HBP is considered to be an emerging health issue in children and adolescents [2,3,4], and it tends to become a global problem. Data are scarce on the incidence of HTN in childhood; however, it is estimated that the prevalence rate varies from 1% to 5% [5,6]. A recent study in Poland [7] indicated that the prevalence of HTN ranged from 5.6% to 7.9% in 10–18-year-old adolescents. The same study found a relationship between HTN and low socioeconomic status, living in a rural environment and parents’ education. Children and adolescents with primary HTN are frequently overweight or obese [8]. As Body Mass Index (BMI) is increasing, the prevalence of HTN is increased progressively. High values of BP in childhood are connected to target organ damage, which couldlead to premature cardiovascular morbidity and mortality in adults, and commonly leads to HTN in young adulthood [9]. According to the working group from national high blood pressure education program that investigated high blood pressure in children and adolescents [10], diagnosis of HTN is complicated, because BP values vary with age, gender and height. Recently, Lu et al. [11] developed a novel criterion, using ratios of systolic BP to height and diastolic BP to height to identify HBP. Since then, this new criterion has been used in several studies in the world [12,13,14,15] demonstrating high specificity and sensitivity from cutoff points determined by receiver operating characteristic (ROC) curves. 

A recent survey in Serbia [16], which included 14,623 adult respondents, found that one in three adults had prehypertension (33.1%) and every other hypertension (49.3%). Such prevalence places Serbia in the countries with high prevalence of prehypertension/hypertension. Due to the fact that HTN in children and adolescents can lead to adult HTN, appropriate early-stage diagnosis and intervention in children and adolescents are important for reducing the risk of HTN-related disorders in adults [17].

This study aimed to compare BMI-for-age (Z score) and blood pressure-to-height ratio (BPHR) in predicting incidence of hypertension, and for the first time to evaluate the diagnostic value of BPHR for diagnosing hypertension in children and adolescents from the north part of Serbia.

## 2. Materials and Methods

### 2.1. Participants 

The cross-sectional study was performed in the period between 2017 and 2019 and included 2287 children (<13 years) and adolescents (≥13 years). In total, 1133 male and 1154 female subjects aged 7–15 years were investigated. The data used in this study were gathered in a large-scale survey of children and adolescents, covering most of the geographical areas of the North Bačka region of Vojvodina (Republic of Serbia).

### 2.2. Procedures

Age was calculated as the difference between the date of birth and date of data collection. Each age group was categorized according to the midpoint of an age range. For example, the group of participants aged 7 years included all participants between 6.50 and 7.49 years. The subjects were grouped into nine age categories (6.50–15.49). Informed consent was obtained from participants and their parents before data collection, and the inclusion of subjects was on a voluntary basis. The written consent for conducting the surveys was also obtained from the Academic Council of Faculty of Sciences and Department of Biology and Ecology in Novi Sad, the Provincial Secretariat for Higher Education and Scientific Research and primary and secondary school principals. The surveyed traits included height and weight. Height was measured with an anthropometer (±1 mm; SieberHegnerMaschinen AG, Zürich, Switzerland) with the head positioned in the Frankfurt plane, and a portable electronic digital scale was used to measure weight with accuracy of ±0.1 kg. Blood pressure (BP) was measured three times, with digital sphygmomanometer OMRON M6 Comfort from the bare right arm of participants, at the level of the heart, while participants remained seated with back support, after at least 5 min of rest.

Body mass index (BMI) was calculated as weight (kg) divided by height squared (m^2^). The subjects were classified into four groups: underweight, normal, overweight and obese, according to the International Obesity Task Force—IOTF [18]. The following equations (1) and (2) according to 11 and 12 for BPHR were used:systolic BPHR (SBPHR) = SBP (mm Hg)/height (cm),(1)
diastolic BPHR (DBPHR) = DBP (mm Hg)/height (cm),(2)

To assess blood pressure categories, we used the criterion suggested by the European Society of Hypertension [19] and which refers to children <16 years: normal <90th percentile; elevated ≥90–<95th percentile; hypertension ≥95th percentile. All children who had blood pressure ≥95th percentile were categorized as having hypertension, with no categorization for stage 1 hypertension and stage 2 hypertension. The same blood pressure range norms for non-overweight and overweight children were used. No data on the existence of possible target organ damage were taken from the children.

### 2.3. Statistical Analysis

In order to determine the accuracy of SBPHR, DBPHR and BMI as diagnostic tests for elevated blood pressure (elevated BP), we used the receiveroperating characteristic curve analyses. The discriminating power of the BMI and the BPHR was expressed as area under the curve (AUC) and 95% confidence intervals (CI). An AUC value of ≥0.90 was considered an excellent accuracy; 0.80–0.89 good; 0.70–0.79 satisfactory and <0.70 bad accuracy. The sensitivity and specificity of BPHR and BMI as indicators of HTN were determined with cutoff values. The Youden index was used to determine optimal cutoff values BMI and BPHR for identification of elevated BP/HTN (maximum value of (sensitivity + specificity − 1)) [20].

This study was approved by the Institutional Review Committee of the University of Novi Sad, Provincial Secretariat for Higher Education and Scientific Research (Ref. No. 128-611-68/2019-01), and was conducted under the Declaration of Helsinki.

## 3. Results

The general characteristics of the participants are shown in Table 1. Mean height, BMI (*p* < 0.01), weight and SBP (*p* < 0.001) were higher in boys, while DBP was higher in girls (*p* < 0.05).

Based on the IOTF categorization (Table 2), it can be noticed that the prevalence of overweight is 18.3% and obesity is present in 8.8% of adolescents. Obesity is more prevalent in boys (*p* = 0.049), while underweight is more prevalent in girls (*p* = 0.000). The prevalence of elevated BP and HTN shows no statistically significant differences between boys and girls. Elevated SBP and DBP in general resulted in a prevalence of 6.6%, and for the HTN the prevalence is 5.05%. The correlation between BMI-for-age (Z score) and blood pressure was significant and positive for SBP (*r* = 0.300 for boys; *r* = 0.302 for girls), while there was no correlation for DBP (*r* = 0.092 for boys; *r* = 0.006 for girls). 

The areas under the ROC curve of BMI-for-age (Z score), SBPHR and DBPHR as predictors of elevated BP and HTN in boys and girls, and their confidence intervals (95% CI), as well as the cutoff points and their relevant sensitivity and specificity, can be seen in Table 3.

It is noted that both BMI-for-age (Z score) and BPHR take over larger areas, but BMI-for-age Z-score has a modest ability and, on the contrary, BPHR shows very good ability to identify children with elevated BP and HTN. The optimal thresholds of SBPHR/DBPHR for defining elevated BP were 0.79/0.55 for boys and 0.80/0.54 for girls, and for defining HTN were 0.83/0.57 for boys and 0.82/0.54 for girls. The sensitivity ranged from 78.5% to 95.7%, and the specificity from 73.9% to 87.6%.

## 4. Discussion

This study aimed to identify the predictive power and to propose cutoff points of BMI-for-age (Z score) and blood pressure-to-height ratio (BPHR) for prediction of high blood pressure in school children of Vojvodina (North Serbia). It is difficult to compare the prevalence of HTN in children and adolescents due to methodological differences and different reference criteria which can cause high variability in the prevalence of high blood pressure. In this study, the prevalence of elevated BP was 6.6% and for the HTN the prevalence was 5.05%. These results are similar with recent investigations in Serbia [21] which established a prevalence of high blood pressure in 4.9% of boys and 3.6% of girls of the same age. Our results are slightly higher than the estimated prevalence of clinical hypertension in children and adolescents, which is ~3.5% [5,22], and the prevalence of elevated BP, which is ~2.2% to 3.5%, with higher rates among children and adolescents who are overweight and have obesity [5,23]. According to the new American Academy of Pediatrics guideline [24], hypertension remains at 2% to 4% in 10–17-year-old children in America. However, the prevalence obtained in our study is in agreement with the prevalence of 5% given to children by the Fourth Report on the Diagnosis, Evaluation, and Treatment of High Blood Pressure in Children and Adolescents [10]. The results are also in line with the investigations in Poland [25], where the estimated prevalence was about 6% in children between 3 and 18 years. The prevalence of overall elevated BP/hypertension in this study defined by values above the 90th percentile is in agreement with the results obtained in the USA [26], where the prevalence was higher than 10% among children between 8 and 17 years. However, a large comprehensive survey of children aged 6–17 in China [27] found significantly higher prevalence for elevated BP (13.41%) and HTN (18.25%). 

Hypertension in childhood may remain undiagnosed, and the reasons are mostly related to the measurement and interpretation of BP, as well as complicated diagnostic criteria. In this study, we have shown that BMI-for-age (Z-score) has a modest ability to identify children with elevated BP/HTN with AUC values ranging from 0.625 to 0.723 with low percentages of sensitivity (62% to 72.5%) and specificity (58.2% to 67.3%). On the contrary, BPHR had excellent predictive ability to identify elevated BP/HTN with AUC values of 0.882 to 0.949 for SBP and 0.777 to 0.904 for DBP. Sensitivity ranged from 78.5% to 95.7% and specificity from 73.9% to 87.6%. This study showed that BPHR ratios are a simple and accurate method of diagnosing high values of blood pressure. The findings are in agreement with the previous studies on this matter [8,11,12,13,14,15,28], which demonstrated the feasibility and accuracy of the SBPHR and DBPHR for screening elevated BP/HTN. The optimal cutoffs of SBPHR for defining elevated BP were 0.79 for boys and 0.80 for girls, and for defining HTN were 0.83 for boys and 0.82 for girls. The cutoffs of DBPHR for defining elevated BP were 0.55 for boys and 0.54 for girls, and for defining HTN were 0.57 and 0.54, for boys and girls, respectively.

The results of our study are in line with other investigations as they show that BPHR is a reliable tool for diagnosing and screening for high blood pressure, and can be easily used to screen children at high risk of elevated BP/HTN in large epidemiologic surveys or by children and their parents. However, the limitations of this research lie in that the subcategories of hypertension, i.e., stage 1 and stage 2, were not separately considered. Also, data on the existence of possible target organ damage were not taken into account, which would be helpful to reinforce the use of the indexation of BP over height. However, the contribution of this study is that it is based on a large number of participants, and the results are in line with other investigations, both in Serbia and other regions of the World. It shows that BPHR is a reliable tool for diagnosing and screening for high blood pressure, and can be easily used to screen children at high risk of elevated BP/HTN in large epidemiologic surveys or by children and their parents. Nevertheless, these methods can only be used as screening tools for the identification of children and adolescents who need further evaluation of their BP. They should not be used to diagnose elevated BP or HTN by itself. The screened children with potential hypertension should be assessed by medical professionals.

## 5. Conclusions

BPHR is a simple, screening and diagnostic tool with high sensitivity and specificity to detect elevated BP. To the best of our knowledge, this study is the first in the North part of Serbia to report the predictive ability of BPHR for elevated BP/HTN, which can be used as a reliable tool for diagnosing and also screening for high BP in ambulatory care setting in children.

## Figures and Tables

**Table 1 children-07-00254-t001:** The general characteristics of the participants.

Age		Height (cm)		Weight (kg)		BMI (kg/m^2^)		SBP (mmHg)		DBP (mmHg)	
(years)	*n*	Mean	SD	Mean	SD	Mean	SD	Mean	SD	Mean	SD
Boys											
7	84	126.3 ^a^	5.8	28.8 ^b^	8.2	17.9 ^a^	3.8	100.8	13.2	69.8	9.8
8	160	129.2	6.0	29.2	6.2	17.4	2.7	100.3	12.6	69.8	8.6
9	158	135.6 ^a^	5.9	33.6 ^a^	6.6	18.2	2.8	104.4	11.3	72.0	7.6
10	161	141.4	7.3	39.3 ^b^	11.6	19.4 ^b^	4.6	107.1 ^b^	12.0	73.3	9.1
11	161	146.7	7.3	42.6	10.5	19.6	3.8	109.4	12.1	71.7	8.0
12	148	153.0	8.8	46.3	12.8	19.6	4.1	110.6	11.5	71.9	8.5
13	117	158.7	10.0	51.8	13.8	20.4	4.5	114.9	11.7	71.1	7.7
14	99	168.2 ^c^	8.7	60.9 ^c^	14.2	21.4	4.1	119.2 ^c^	11.4	71.0	7.7
15	45	171.7 ^c^	9.6	63.9 ^c^	14.1	21.5	3.5	122.4 ^b^	10.7	70.8	8.7
Total	1133	145.4 ^b^	15.4	41.8 ^c^	15.0	19.2 ^b^	4.0	108.6 ^c^	13.4	71.3	8.3
Girls											
7	85	124.6	4.9	26.0	5.3	16.7	2.6	99.8	8.9	72.9 ^a^	6.4
8	170	129.9	5.9	29.2	6.7	17.2	3.0	101.0	10.9	72.2 ^a^	7.7
9	163	133.8	6.7	31.8	7.3	17.6	3.0	102.0	11.8	72.0	9.1
10	169	141.4	6.9	36.4	8.0	18.1	3.2	103.5	11.8	72.1	7.8
11	209	147.5	7.2	42.5	10.4	19.4	3.8	107.1	12.0	72.5	8.7
12	142	153.3	8.0	46.8	10.8	19.7	3.4	110.8	12.5	72.6	8.3
13	97	157.8	6.2	52.6	11.9	21.0	4.0	113.4	12.5	72.3	7.8
14	88	162.3	5.7	55.4	8.8	21.1	3.3	112.6	10.8	72.7	8.6
15	31	164.0	5.3	57.9	10.8	21.5	3.7	115.4	11.6	69.3	7.4
Total	1154	143.6	13.5	39.7	13.0	18.8	3.6	106.2	12.5	72.1 ^a^	8.1

Independent-Samples T-test: ^a^
*p* < 0.05; ^b^
*p* < 0.01; ^c^
*p* < 0.001; SD–Standard deviation; BMI–Body mass index; SBP–Systolic Blood Pressure; DBP–Diastolic Blood Pressure.

**Table 2 children-07-00254-t002:** Prevalence of BMI categories and elevated blood pressure.

		Boys		Girls		All		*p*-Value ^#^
		*n*	%	*n*	%	*n*	%	
BMI (kg/m^2^)	UW	42	3.7	82	7.1	124	5.4	0.000 *
	NW	754	66.5	789	68.4	1543	67.5	0.373
	OW	222	19.6	196	17.0	418	18.3	0.273
	OB	115	10.2	87	7.5	202	8.8	0.049 *
SBP (mmHg)	P90 > SBP < P95	79	7.0	69	6.0	148	6.5	0.411
	SBP ≥ P95	45	4.0	47	4.1	92	4.0	0.835
DBP(mmHg)	P9 > DBP < P95	67	5.9	87	7.5	154	6.7	0.107
	DBP ≥ P95	69	6.1	70	6.1	139	6.1	0.932

^#^ Chi–square test; * Statistical significant at *p* < 0.05; BMI–Body Mass Index; UW–underweight; NW–normal; OW–overweight; OB–obese; SBP–Systolic Blood Pressure; DBP–Diastolic Blood Pressure.

**Table 3 children-07-00254-t003:** Area under ROC curve, 95% CI, cutoff points, sensitivity and specificity between BMI-for-age (Z-score), BPHR and high blood pressure.

Risk Factor	Measure	AUC	95% CI	Cutoff	Sen	Spec
Boys						
Systolic BP ≥ P90	BMI-for-age (Z-score)	0.674	0.610–0.737	0.130	0.620	0.667
	SBPHR (mmHg/cm)	0.836	0.796–0.875	0.790	0.785	0.739
Diastolic BP ≥ P90	BMI-for-age (Z-score)	0.506	0.448–0.578	−1.609	0.462	0.542
	DBPHR (mmHg/cm)	0.777	0.730–0.825	0.545	0.677	0.761
Systolic BP ≥ P95	BMI-for-age (Z-score)	0.625	0.546–0.704	−0.070	0.644	0.582
	SBPHR (mmHg/cm)	0.935	0.908–0.962	0.830	0.911	0.855
Diastolic BP ≥ P95	BMI-for-age (Z-score)	0.502	0.421–0.582	−0.822	0.479	0.573
	DBPHR (mmHg/cm)	0.904	0.871–0.937	0.577	0.746	0.881
Girls						
Systolic BP ≥ P90	BMI-for-age (Z-score)	0.646	0.577–0.714	−1.930	0.725	0.506
	SBPHR (mmHg/cm)	0.882	0.856–0.909	0.800	0.884	0.808
Diastolic BP ≥ P90	BMI-for-age (Z-score)	0.503	0.436–0.569	−2.198	0.506	0.479
	DBPHR (mmHg/cm)	0.782	0.741–0.824	0.535	0.732	0.678
Systolic BP ≥ P95	BMI-for-age (Z-score)	0.723	0.643–0.803	0.220	0.681	0.673
	SBPHR (mmHg/cm)	0.949	0.931–0.968	0.820	0.957	0.876
Diastolic BP ≥ P95	BMI-for-age (Z-score)	0.476	0.403–0.548	−0.096	0.453	0.575
	DBPHR (mmHg/cm)	0.857	0.813–0.900	0.542	0.813	0.715

P90–90th percentile; P95–95th percentile; ROC–receiver operating characteristic; AUC–area under the ROC Curve; 95% CI–confidence interval at 95%; Sen–Sensitivity; Spec–Specificity; BP-Blood Pressure; BMI–Body Mass Index; SBPHR-Systolic Blood Pressure-to-Height Ratio; DBPHR–Diastolic Blood Pressure-to-Height Ratio.

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
