# Peer review of "Body Mass Index and Blood Pressure-to-Height Ratio in Predicting Incidence of Hypertension in Serbian Children"

_children, 2020, doi:10.3390/children7120254_

Round 1
Reviewer 1 Report
Authors aimed to assess the diagnosis of hypertension in Serbian children according to 2 different definitions: the AAP, using 2 different cutoff values for the age < 13 and > 13 years and by indexing BP to height
Major comments
Being Serbia a country in Europe, I would suggest to use the nomograms proposed by the ESH Guidelines (Stabouli S, Redon J, Lurbe E. Redefining hypertension in children and adolescents: a review of the evidence considered by the European Society of Hypertension and American Academy of Pediatrics guidelines. J Hypertens. 2020;1:1. https://doi. org/10.1097/hjh.0000000000002247) for the definition of hypertension.
Authors should compare the indexation of BPs over height to the values indicated by ESH normograms, in addition to the APP criteria
Do authors have information about some target organ damage in these children /adolescents ?
This would be helpful to reinforce the use of the indexation of BP over height, if more strictly related to the presence of TOD.
Author Response
Re: Children-976166
To the Editors
Dear Editor,
Please find attached the revised version of our manuscript children-976166, entitled “Body Mass Index and Blood Pressure-to-Height Ratio in Predicting Incidence of Hypertension in Serbian children”
We thank the reviewers for their careful evaluation and helpful comments to our manuscript. We have carefully taken their comments into consideration in preparing our revision, which has resulted in a paper that is clearer, broader and more compelling.
Some changes were made to this paper in order to improve it. Please find below our point-by-point responses to each of the comments of the reviewers. All changes are marked in the text.
We hope that the revisions in the manuscript and our accompanying responses will be sufficient to make our manuscript suitable for publication in Children.
Sincerely,
Maja Batez
Authors aimed to assess the diagnosis of hypertension in Serbian children according to 2 different definitions: the AAP, using 2 different cutoff values for the age < 13 and > 13 years and by indexing BP to height
Our response: Thank you for your detailed review of our manuscript and for providing some insightful and thought-provoking suggestions to strengthen our manuscript. We feel we have sufficient responses to each of your major concerns listed above, which are further detailed below, and hope that they alleviate the concerns you have regarding the approaches adopted in our manuscript.
Major comments
Being Serbia a country in Europe, I would suggest to use the nomograms proposed by the ESH Guidelines (Stabouli S, Redon J, Lurbe E. Redefining hypertension in children and adolescents: a review of the evidence considered by the European Society of Hypertension and American Academy of Pediatrics guidelines. J Hypertens. 2020;1:1. https://doi. org/10.1097/hjh.0000000000002247) for the definition of hypertension.
Our response: We used the criterion according to European Society of Hypertension which is valid for children <16 years, and which was suggested. Thank you for your suggestion.
Authors should compare the indexation of BPs over height to the values indicated by ESH normograms, in addition to the APP criteria
Our response: We apologize for the misunderstanding. In the first version, it is written that the subjects were divided into two groups <13 years (children) and> 13 years (adolescents), thinking only of the age of the examined children, and not according to the APP criterion for assessing blood pressure categories. The definition of blood pressure categories has been done according to the European categorization from the beginning. In this paper, we did not use the categorization proposed by the APP criterion but the categorization that is given by the ESH (European Society of Hipertension).
Do authors have information about some target organ damage in these children /adolescents ?
This would be helpful to reinforce the use of the indexation of BP over height, if more strictly related to the presence of TOD.
Our response: Thank you for this comment and pointing out the importance of target organ damage in children /adolescents for the indexation of BP over height. However, this information was not available to us and therefore we have mention this as a limitation of our study.

Reviewer 2 Report
I have two main comments.
The Authors use two citation styles (citation using author’s surname and citation in brackets) – this thing should be corrected and unified.
However my main concern refers to methods used by the Authors. The Authors defined hypertension as the value >=95 percentile. The Authors should point the source of percentile table (the source of the range norms) they used to classify the patients as the ones with/without hypertension. What is more, the Authors emphasised the problem of overweight and obese children among the group. Did Authors used the same blood pressure range norms for non-overweight and overweight patients?
The presented results are align with previous publications in this area. As the hypertension diagnosis in children is based on three factors: BP value, height and age, is was pretty obvious that BPHR - the index including two of them (BP value and height) is going to be better that BMI to predict the presence of hypertension. However apart from this rather low novelty, the Authors set the cut-off point for BPHR that can be helpful in screening but not in clinical practice, with is the practical value of the paper.
Author Response
November 14, 2020
Re: Children-976166
To the Editors
Dear Editor,
Please find attached the revised version of our manuscript children-976166, entitled “Body Mass Index and Blood Pressure-to-Height Ratio in Predicting Incidence of Hypertension in Serbian children”
We thank the reviewers for their careful evaluation and helpful comments to our manuscript. We have carefully taken their comments into consideration in preparing our revision, which has resulted in a paper that is clearer, broader and more compelling.
Some changes were made to this paper in order to improve it. Please find below our point-by-point responses to each of the comments of the reviewers. All changes are marked in the text.
We hope that the revisions in the manuscript and our accompanying responses will be sufficient to make our manuscript suitable for publication in Children.
Sincerely,
Maja Batez
Reviewer 2
We wish to thank you for your constructive comments in this review. Your comments provided valuable insights to refine its contents and analysis. In this document, we try to address the issues raised as best as possible. Thank you again for giving us the opportunity to submit a revised draft of our manuscript.
I have two main comments.
The Authors use two citation styles (citation using author’s surname and citation in brackets) – this thing should be corrected and unified.
Our response: We apologize for the mistakes and inconsistency, we have revised the references.
However my main concern refers to methods used by the Authors. The Authors defined hypertension as the value >=95 percentile. The Authors should point the source of percentile table (the source of the range norms) they used to classify the patients as the ones with/without hypertension. What is more, the Authors emphasised the problem of overweight and obese children among the group. Did Authors used the same blood pressure range norms for non-overweight and overweight patients?
Our response: We are sorry for omitting this important information. We have added the reference in that was used for the classification of the participants to one with or without hypertension. The change can be seen on the third page in Materials and Methods section.
We also indicated that the same blood pressure range norms for non-overweight and overweight children were used.
The presented results are align with previous publications in this area. As the hypertension diagnosis in children is based on three factors: BP value, height and age, is was pretty obvious that BPHR - the index including two of them (BP value and height) is going to be better that BMI to predict the presence of hypertension. However apart from this rather low novelty, the Authors set the cut-off point for BPHR that can be helpful in screening but not in clinical practice, with is the practical value of the paper.
Our response: We really appreciate your time and effort to make this paper better and soundable for readers. We found that the BPHR is a strong predictor of hypertension in Serbian populations and have advantages as a screening measurement tool, compared to BMI. Although without novelty, we think it is important that Intensive lifestyle modifications should be introduced early in order to reduce the risk of hypertension.
indings of our study suggest that the baseline WHtR
is a strong predictor of hypertension in Korean popula-
tions. In addition, the WHtR was found to have several
advantages as a screening measurement tool, compared to
BMI. Intensive lifestyle modifications should be intro-
duced early to reduce the WC to less than half of the
height in order to reduce the risk of hypertension

Round 2
Reviewer 1 Report
authors have addressed all major issues raised by this reviewer . In particular they have used the european criteria for the definition of children hypertension
Reviewer 2 Report
The Authors revised and improved the manuscript significantly. All confusions have been explained.
Although the study in general does not bring much novelty (which I already expressed in my previous review), it is properly performed and described. I have no other comments.